# Degradation of arouser by endosomal microautophagy is essential for adaptation to starvation in *Drosophila*

Anne-Claire Jacomin[1] , Raksha Gohel[1,*], Zunoon Hussain[1,*], Agnes Varga[2] , Tamas Maruzs[3], Mark Eddison[4], Margaux Sica[5], Ashish Jain[9,10,11] , Kevin G Moffat[1], Terje Johansen[9] , Andreas Jenny[5,6,7,8] , Gabor Juhasz[2,3] , Ioannis P Nezis[1]

**Hunger drives food-seeking behaviour and controls adaptation of organisms to nutrient availability and energy stores. Lipids constitute an essential source of energy in the cell that can be mobilised during fasting by autophagy. Selective degradation of proteins by autophagy is made possible essentially by the presence of LIR and KFERQ-like motifs. Using in silico screening of *Drosophila* proteins that contain KFERQ-like motifs, we identified and characterized the adaptor protein Arouser, which functions to regulate fat storage and mobilisation and is essential during periods of food deprivation. We show that hypomorphic *arouser* mutants are not satiated, are more sensitive to food deprivation, and are more aggressive, suggesting an essential role for Arouser in the coordination of metabolism and food-related behaviour. Our analysis shows that Arouser functions in the fat body through nutrient-related signalling pathways and is degraded by endosomal microautophagy. Arouser degradation occurs during feeding conditions, whereas its stabilisation during non-feeding periods is essential for resistance to starvation and survival. In summary, our data describe a novel role for endosomal microautophagy in energy homeostasis, by the degradation of the signalling regulatory protein Arouser.**

## Introduction

The coordination of metabolism and feeding behaviour according to nutrient availability is crucial to maintain organismal homeostasis, fitness and survival. In many animals during feeding excess nutrients such as carbohydrates are converted into lipids, mainly in the form triacylglycerols (TAGs), through a process called lipogenesis. Lipid breakdown via lipolysis constitutes an essential source of energy when sugars are not readily available (Zechner et al, 2017). The TAGs are stored as lipid droplets in adipose tissue. In *Drosophila*, the main adipose tissue is called the fat body (Rajan & Perrimon, 2013).

In mammals, the balance between lipogenesis and lipolysis is regulated by the nutrient sensing pathways of the mTOR complex (Cai et al, 2016). When nutritionally satiated, activation of mTOR triggers signalling cascades that, in turn, lead to the activation of SREBP transcription factors; the "master regulators" of lipogenic genes. Insulin signalling is well established in regulating lipid metabolism and is well-conserved in all animals, including *Drosophila* (Saltiel & Kahn, 2001; Garofalo, 2002; Kannan & Fridell, 2013). Activation of mTOR due to overfeeding causes a severe down-regulation of autophagy, leading to insulin resistance. Long-term overactivation of the mTOR complex activates lipogenesis and thus contributes to obesity (Hotamisligil, 2010; Yang et al, 2010; Xu et al, 2016). Insulin-mediated mTOR signalling controls both autophagy and lysosome functions (Puertollano, 2014).

Autophagy is a catabolic process responsible for the degradation of intracellular components in lysosomes. Three types of autophagy have been described: macroautophagy, chaperone-mediated autophagy (CMA) and endosomal microautophagy (eMi). Macroautophagy implies the isolation of cytoplasmic components into autophagosomes that eventually fuse with lysosomes. CMA relies on the translocation of cytosolic substrates directly into the lysosome via the lysosomal-associated membrane protein 2A (LAMP2A) (Cuervo & Dice, 1996). Finally, eMi consists of the internalisation of substrates through invagination of the lysosomal or late endosomal membranes (Sahu et al, 2011). Both CMA and eMi rely on the recognition of KFERQ

[1]School of Life Sciences, University of Warwick, Coventry, UK   [2]Department of Anatomy, Cell and Developmental Biology, Eotvos Lorand University, Budapest, Hungary   [3]Institute of Genetics, Biological Research Centre, Szeged, Hungary   [4]Department of Anatomy, University of California San Francisco, San Francisco, CA, USA   [5]Department of Developmental and Molecular Biology, Albert Einstein College of Medicine, New York, NY, USA   [6]Institute for Aging Studies, Albert Einstein College of Medicine, New York, NY, USA   [7]Marion Bessin Liver Research Center, Albert Einstein College of Medicine, New York, NY, USA   [8]Department of Genetics, Albert Einstein College of Medicine, New York, NY, USA   [9]Molecular Cancer Research Group, Institute of Medical Biology, University of Tromsø–The Arctic University of Norway, Tromsø, Norway   [10]Centre for Cancer Cell Reprogramming, Institute of Clinical Medicine, Faculty of Medicine, University of Oslo, Oslo, Norway   [11]Department of Molecular Cell Biology, Institute for Cancer Research, Oslo University Hospital, Oslo, Norway

Correspondence: I.Nezis@warwick.ac.uk
*Raksha Gohel and Zunoon Hussain contributed equally to this work
Anne-Claire Jacomin and Ioannis P Nezis are equal senior authors

motif-containing cargos by the chaperone protein Hsc70/HSPA8, although the latter can also occur in bulk (Dice, 1990; Sahu et al, 2011; Tekirdag & Cuervo, 2018). Because of the lack of LAMP2A homologue in *Drosophila*, it has been suggested that eMi and chaperone-associated selective autophagy are the only types of autophagy that can depend on the Hsc70 chaperone homologue Hsc70-4 (Arndt et al, 2010; Uytterhoeven et al, 2015; Mukherjee et al, 2016). Endosomal microautophagy (eMi) and CMA are known to be activated by prolonged starvation (Cuervo et al, 1995; Ferreira et al, 2015; Mukherjee et al, 2016), eMi was also shown to allow for rapid degradation of selective autophagy receptors upon starvation (Mejlvang et al, 2018). Nonetheless, basal CMA activity was reported, allowing for the degradation of AF1Q protein in mammalian cells (Li et al, 2014). Autophagy has been traditionally linked to cellular energy balance and to the cellular nutritional status. Indeed, early studies showed that activation of autophagy during starvation is necessary to maintain the energetic balance of the cell (Singh & Cuervo, 2011, 2012). These studies emphasised the ability of autophagy to supply free amino acids through the lysosomal degradation of unnecessary proteins to maintain the synthesis of essential proteins under extreme nutritional conditions. Recent studies have shown that autophagy can provide energetically more efficient components from the degradation of lipids (Singh & Cuervo, 2012). Whereas the degradation of lipids and lipid droplet-associated proteins by autophagy has been described, the selective degradation of signalling molecules that regulate lipid homeostasis remains largely unknown.

In the present study, we identify Arouser, a predicted adaptor protein containing PTB (phosphotyrosine binding) and SH3 domains, as a novel substrate for eMi in *Drosophila*. Degradation of Arouser occurs during feeding states of the animal, whereas its stabilisation during non-feeding is essential for resistance to starvation and survival. We show that Arouser is associated with the lysosome and contributes to the regulation of lipid metabolism through regulation of insulin signalling.

# Results

### An in silico, proteome-wide analysis, for KFERQ-like motifs in *Drosophila*

To identify novel proteins involved in eMi, we screened the *Drosophila* proteome for KFERQ-like motifs which are representative of eMi substrates (Uytterhoeven et al, 2015; Mukherjee et al, 2016). An in silico gene ontology approach was conducted using the PANTHER GO-SLiM classification system for both [KR][FILV][DE][KRFILV]Q and Q[KR][FILV][DE][KRFILV] motifs (Dice, 1990). The first motif, [KR][FILV][DE][KRFILV]Q, is referred to as motif 1 and Q[KR][FILV][DE][KRFILV] as motif 2. Proteins from *Drosophila* proteome containing KFERQ-like motifs were obtained using the web-resource SLiMSearch (Krystkowiak & Davey, 2017). A total of 727 entries were identified for motif 1, and 932 for motif 2 (Table S1). The characterised *Comatose/Comt* gene, a known eMi substrate in *Drosophila*, was identified. Biological process analysis found an array of categories associated with KFERQ-like motif-containing proteins. For both motifs, the most common processes were biological regulation, cellular

processes, and metabolic processes (Fig S1A and E). These processes are fairly extensive, indicating that eMi substrates may have broad functions and that eMi controls a wide range of biological processes in *Drosophila*. Analysis of cellular components of KFERQ-like motif-containing proteins found almost identical profiles for both motifs. Genes that encoded proteins in the cell were the most abundant, as well as those in protein complexes (Fig S1B and F). This suggests that eMi regulates proteins essential in the structure and function of the cell, as well as regulating the formation and function of protein complexes. Molecular functions were analysed for both KFERQ-like motifs which found that the highest category for both was binding activity, closely followed by catalytic activity (Fig S1C and G). Protein classes were also investigated, finding nucleic acid–binding and enzyme modulator proteins as two of the highest hits for both motifs (Fig S1D and H). This suggests that eMi may control activities such as transcription, DNA repair, RNA splicing, and replication. The motifs differ slightly in their protein class hits, for example, motif 1 has higher abundance of genes encoding hydrolases, whereas motif 2 is more abundant in cytoskeletal proteins (Fig S1D and H). There are many different protein classes for each motif and, therefore, each class has a relatively low percentage of gene hits (<10%) (Fig S1D and H). This implies that KFERQ-like motif-containing proteins cover a spectrum of protein classes and therefore eMi may not be specific to any explicit classes in particular.

One of our hits was the predicted adaptor protein Arouser, containing PTB and SH3 domains, which contains a KFERQ-like motif at position 303-307 (RLEVQ) (Fig 1A) (Tocchetti et al, 2003; Offenhauser et al, 2006). Previous reports showed that Arouser is involved in the regulation of ethanol sensitivity and memory in *Drosophila* (Eddison et al, 2011; LaFerriere et al, 2011). Arouser was annotated to be associated with signal transduction and hence its regulation by eMi may affect essential cellular functions. The prediction for the presence of a KFERQ-like motif by SLiMSearch was also confirmed using the web-based resource KFERQ Finder (Kirchner et al, 2019). Arouser is a member of the EPS8 (EGF receptor pathway substrate 8) protein family; mammalian EPS8 has been shown to be subject to CMA (Welsch et al, 2010). Therefore, we sought to investigate if this lysosomal degradation of Arouser is conserved in *Drosophila* and how this regulation impact on the animal's physiology.

### Arouser is degraded by endosomal microautophagy

Because Arouser has a putative eMi motif, we examined whether it is localized to late endosomes/lysosomes. We observed that Arouser-GFP colocalises with the late endosomal/lysosomal marker LAMP1-3xmCherry (Figs 1B and S2A–C). To explore if the autophagy-lysosome system regulates Arouser degradation, we fed *Drosophila* larvae for 24 h with the lysosomal inhibitor chloroquine; the well-characterised lysosome-autophagy substrate Ref(2)P was used as a positive control for impairment of the lysosomal activity (Bartlett et al, 2011). We observed that Arouser accumulates in a dose-dependent manner in larvae fed with chloroquine compared with larvae fed on food containing the drug vehicle (Fig 1C and D), suggesting that Arouser is normally targeted to the lysosome for degradation. Consistently, accumulation of Arouser protein was also observed in the mutant larvae *spinster* (*spin*) and *cathepsin D* (*cathD*) that both lack functional lysosomes (Khurana et al, 2010; Rong et al, 2011) (Fig 1E

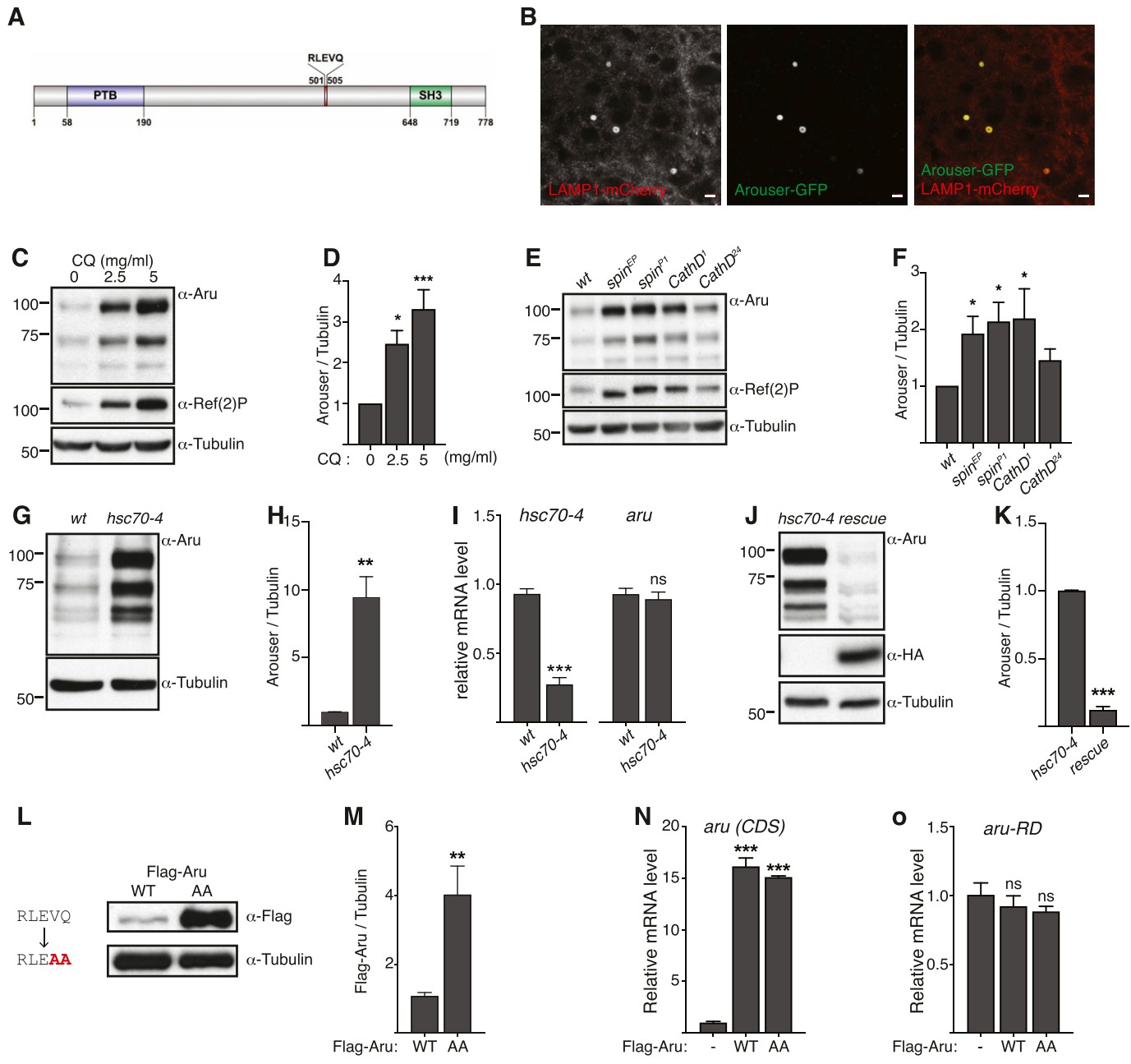

**Figure 1. Arouser is a substrate for endosomal microautophagy.**
**(A)** Domains and motif of Arouser protein. PTB, phosphotyrosine binding domain; SH3, SRC Homology 3 Domain. **(B)** Airyscan confocal section of a fat body cell from starved showing the colocalisation of Arouser-GFP (green) and LAMP1-3xmCherry (red). Scale bar: 2 μm. **(C, D, E, F)** Western blot analysis and quantification of endogenous Arouser protein in larvae fed for 24 h with chloroquine (C, D) or in larvae with defective lysosomes (E, F). **(G, H, I, J, K)** Analysis of endogenous Arouser in larvae with defective eMi (G, H, I) and eMi rescue (J, K); relative gene expression levels for hsc70-4 and aru are shown in (I). **(L, M)** Western blot analysis and quantification of wild-type (WT) and eMi-resistant (AA) Flag-Arouser expressed in the larval fat body. **(N, O)** Relative gene expression for aru pan-isoform (CDS) (N) and endogenous aru using primers specific to aru-RD isoform (O) in flies expressing Flag-Aru wild-type (WT) and eMi mutant (AA); w[1118] flies were used as negative control. Bar charts show means ± s.d. Statistical significance was determined using one-way ANOVA, *$P < 0.05$, **$P < 0.01$, ***$P < 0.001$.

and F). Therefore, we concluded that Arouser is degraded by the lysosome.

In *Drosophila*, endosomal microautophagy allows for the degradation of proteins in the lysosomes and requires the chaperone protein Hsc70-4 (Uytterhoeven et al, 2015), we performed similar analysis of Arouser in *hsc70-4*–deficient larvae. Again, Arouser protein accumulated in these eMi-deficient larvae (Fig 1G and H). Using real-time (RT) qPCR, we showed that the expression level of *aru* gene remained unchanged in *hsc70-4-*–deficient larvae compare with wild-type larvae (Fig 1I). This accumulation of Arouser protein was rescued upon HA-Hsc70-4 rescued expression in an *hsc70-4*–mutant background (Fig 1J and K), as well as in fed larvae

overexpressing a mutated form of Hsc70-4 (HA-Hsc70-4[3KA]) that lacks the ability to deform the membranes (Fig S2I), suggesting that Arouser degradation by eMi is dependent on the chaperone Hsc70-4. This hypothesis of eMi involvement in Arouser degradation was further supported by RNAi-mediated silencing of *Atg1* and *Atg13*, two autophagy-related genes involved in eMi, as well as ESCRT components Stam and Vps25 (Mukherjee et al, 2016). Arouser accumulated in well-fed *Atg1* and *Atg13* silenced larvae compared with a GFP control expressing larvae (Fig S2D–F). Similar observation was made in fed larvae depleted in key components of the ESCRT machinery involved in eMi (Fig S2H). Moreover, we showed that Arouser colocalises with Atg1 (Fig S2G).

Next, we tested the functionality of RLEVQ motif in Arouser. To do this, we created transgenic *UAS-3xFlag-Aru^WT* and *UAS-3xFlag-Aru^AA* flies to overexpress Arouser wild-type (WT) or a mutated (AA) form of the protein where we replaced the two last amino acids of the pentapeptide of the eMi by alanine (VQ > AA). Mutation of the KFERQ-like motif of Arouser resulted in the accumulation of Flag-Aru^AA protein (Fig 1L and M). No difference in the level of transcripts of the ectopic Flag-Aru constructs (*aru* (*CDS*)) (Fig 1N), and from endogenous *aru* gene (*aru-RD*) were observed (Fig 1O). These data suggest that the KFERQ-like motif is an essential motif that targets Arouser for degradation. To test if *aru* mutants are defective in eMi, we used a photoactivable mCherry-KFERQ marker (Mukherjee et al, 2016), and observed that *aru* mutants themselves do not display any eMI defect after 24 h of amino acid starvation in 20% sucrose (Fig S2J–M), suggesting that the loss of Arouser does not disrupt eMi.

Taken together, these results above indicate that Arouser is a novel endosomal microautophagy substrate in *Drosophila*.

### Arouser interacts with Atg8a

Analysis of the Arouser sequence using the iLIR web resource also revealed the presence of putative LIR motifs (Fig S3A), which are required for the selective degradation of most macroautophagy substrates (Kalvari et al., 2014; Johansen & Lamark, 2020). This suggests that Arouser might also be degraded by macroautophagy through Atg8a (Scott et al, 2004; Nezis et al, 2010). Indeed, we observed some colocalisation between overexpressed Arouser-GFP and Atg8a, in larval fat body (Fig S3B). Furthermore, endogenous Arouser co-precipitated with Atg8a in adult flies, suggesting that both proteins can be part of the same complex (Fig S3C). We further tested this interaction between both proteins in vitro and found that Arouser interacts directly with Atg8a. However, the association of Arouser and Atg8a is likely LIR motif-independent because Arouser also interacted with the LIR docking site mutated form of Atg8a (Fig S3D).

To address if Arouser's binding to Atg8a is related to its macroautophagic degradation, we examined if Arouser accumulates in fed *Atg8a*- and *Atg7*-deficient larvae compared with control animals. We did not observe any obvious accumulation of Arouser in *Atg8a*- and *Atg7*-deficient larvae (Fig S3E and F). Instead, Arouser protein levels appear to be reduced in Atg8a-deficient flies; possibly because in the absence of functional macroautophagy, other lysosomal degradative processes—including eMi—are up-regulated. This may suggest that the interaction between Arouser and Atg8a is not probably related to

its degradation by macroautophagy and that Arouser degradation is principally degraded by eMi.

To investigate if Arouser could be involved in the regulation of macroautophagy, we made used of the fluorescent markers mCherry-Atg8a and GFP-mCherry-Atg8a, expressed concomitantly with a RNAi targeting *aru* gene or a control RNAi (Fig S4A–D). No difference in the formation of autophagosomes and autolysosomes was observed in cells silenced for *aru* in fed (Fig S4A and B) or starved (Fig S4C and D) conditions compared with control cells. The ability of the cell to induce the formation of active lysosomes in *aru* mutant larvae was monitored using Lysotracker-Red (LTR) (Fig S4E–J). No difference between the two *aru* mutants and wild-type tissues was observed in fed or starved conditions. These data suggest that Arouser is not involved in the regulation of macroautophagy in *Drosophila* larval fat body cells.

### Arouser is involved in the response to nutrient starvation

Our data support the hypothesis that Arouser is degraded in the lysosome through eMi. Previous work has identified that the autophagy-lysosome pathway is more active under restricted nutrient conditions, contributing to the degradation of unnecessary cellular components to maintain the nutrient load of the cell (Mejlvang et al, 2018; Ravanan et al, 2017). Therefore, we next tested if starvation affects the quantity of Arouser protein, predicting that an increased eMi activity by starvation, will result in decreased Arouser levels. Surprisingly, we noticed that endogenous Arouser protein gradually accumulated in larvae after 4 and 24 h of amino acid starvation in 20% sucrose compared to age-matched well-fed larvae (Fig 2A). The accumulation of a protein may result either from a stabilisation of the protein, which is not degraded anymore, or from an increase in gene expression and protein neo-synthesis. To decipher between these two scenarios, we compared the relative expression level of *aru* between fed and 4-h starved larvae. We observed no significant increase in the expression of *aru* between conditions, suggesting that increased transcription of Arouser does not occur in starved larvae (Fig 2B). GFP-tagged proteins degraded in the lysosomes can show a cleavage of the GFP from the tagged protein (Mauvezin et al, 2014; Klionsky et al, 2016). Using a GFP-tagged Arouser (Aru-GFP) protein ectopically expressed under UAS promoter in the fat body, we observed that Arouser-GFP also accumulated in a time-dependent manner during starvation whilst the cleaved GFP quantity decreased (Fig 2C). Altogether, our results demonstrate that Arouser protein is not targeted for degradation under conditions of nutrient deprivation, suggesting that Arouser may have an important role in the physiological adaptations that occur in response to starvation that ultimately promote survival.

To test this, we evaluated the survival of two *arouser* (*aru*) mutants under conditions of food restriction, using two previously characterized P-element hypomorphic mutations, called *aru^8.128* and *aru^8896* (Eddison et al, 2011). Normally fed 2- to 3-d-old adult males were submitted to either an amino acid deprivation diet, consisting of 5% sucrose, or to a complete starvation on water only agar pads. Both *aru* mutants displayed significantly shorter lifespan than control flies when subjected to food deprivation (Fig 2D and E). We also tested the ability of larvae to overcome acute starvation (water only agar pad) and similarly observed a significant reduction

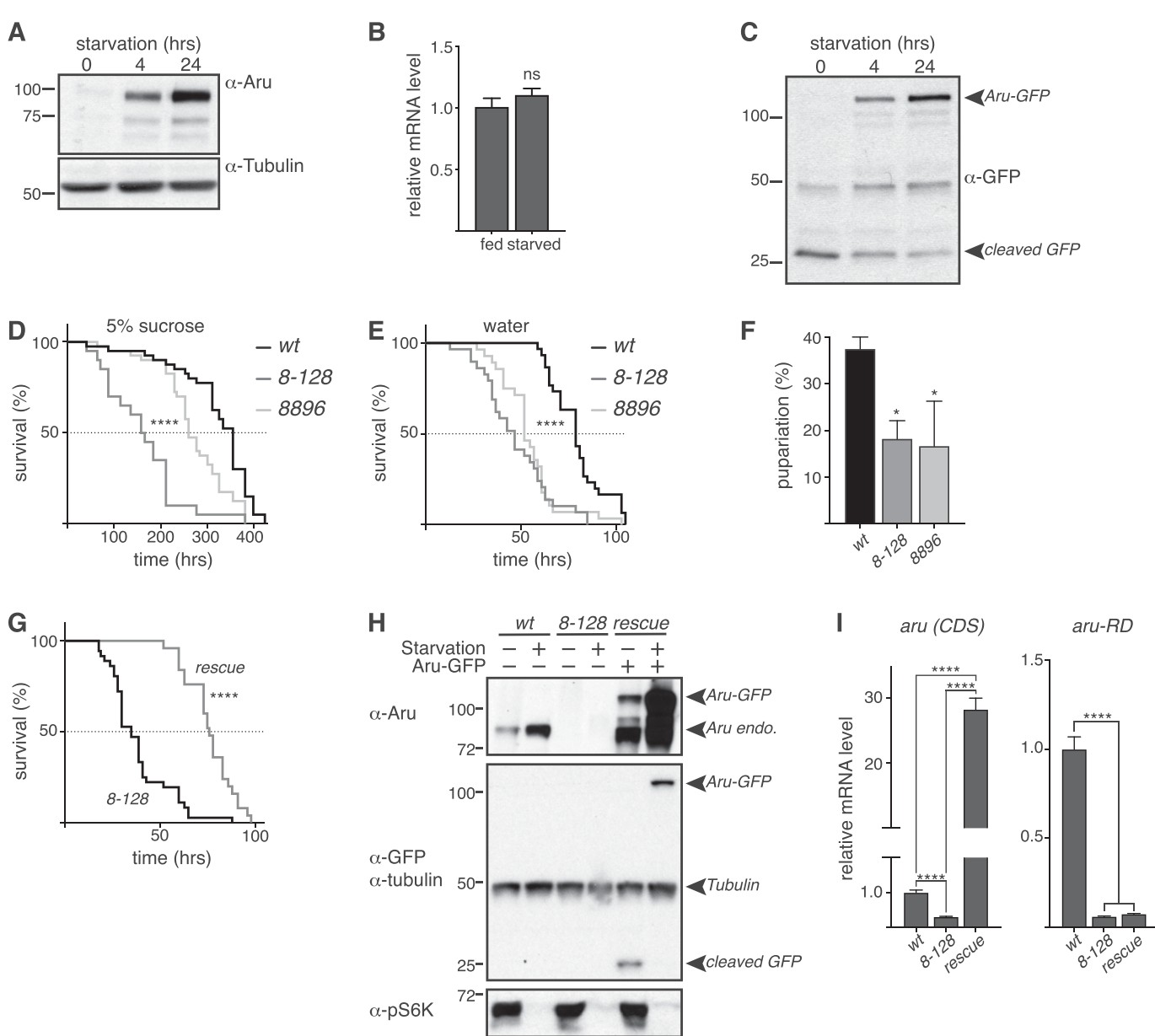

**Figure 2. Arouser is involved in resistance to starvation.**
**(A, C)** Western blot analysis of endogenous (A) and overexpressed (C) Arouser in fed and 4–24 h starved larvae. **(B)** Analysis of *aru* mRNA levels in fed and starved larvae. Bar chart shows means ± s.d. Statistical significance was determined using two-tailed *t* test. **(D, E)** Survival of a 100 wild-type (*w^1118^*) and *aru* mutant males fed on 5% sucrose (D) or water only (E). **(F)** Proportion of pupae form when well-fed second instar larvae are transferred onto water pads. **(G)** Survival on water only of the Arouser rescue line (rescue) compare with the corresponding mutant *aru^8-128^*. **(H)** Western blot validation of the rescue line compared with wild-type and *aru^8-128^* mutant in fed and 4 h starved larvae. **(I)** Relative *aru* gene expression using primers that recognise endogenous and overexpressed *aru* (*CDS*) and primers specific for endogenous arouser (*aru-RD*). Bar chart shows means ± s.d. Statistical significance was determined using one-way ANOVA and based on at least three independent biological replicates, *P < 0.05.

in the proportion of *aru* mutant larvae successfully entering pupariation (Fig 2F). To confirm that the increased sensitivity of the mutants towards food deprivation is specific to the loss of Arouser protein, we made use of the GAL4-containing P[GawB] element inserted in the *aru^8-128^* mutant to express Arouser-GFP. We used Western blotting to confirm that the Arouser-GFP construct was expressed and accumulated during starvation (Fig 2H) and RT-qPCR to verify that the endogenous *aru* gene expression was still absent

in the rescue line (Fig 2I, *aru-RD*) whereas the transgene UAS-Aru-GFP is overexpressed (Fig 2I, *aru*(*CDS*)). We observed that the re-expression of Arouser-GFP in an *aru*-deficient background rescue the resistance of the fly to acute starvation on water (Fig 2G). These data suggest that *aru* is involved in resistance to starvation in both larvae and the adult, suggesting that Arouser may have a crucial role in the adaptation and survival in conditions of food deprivation.

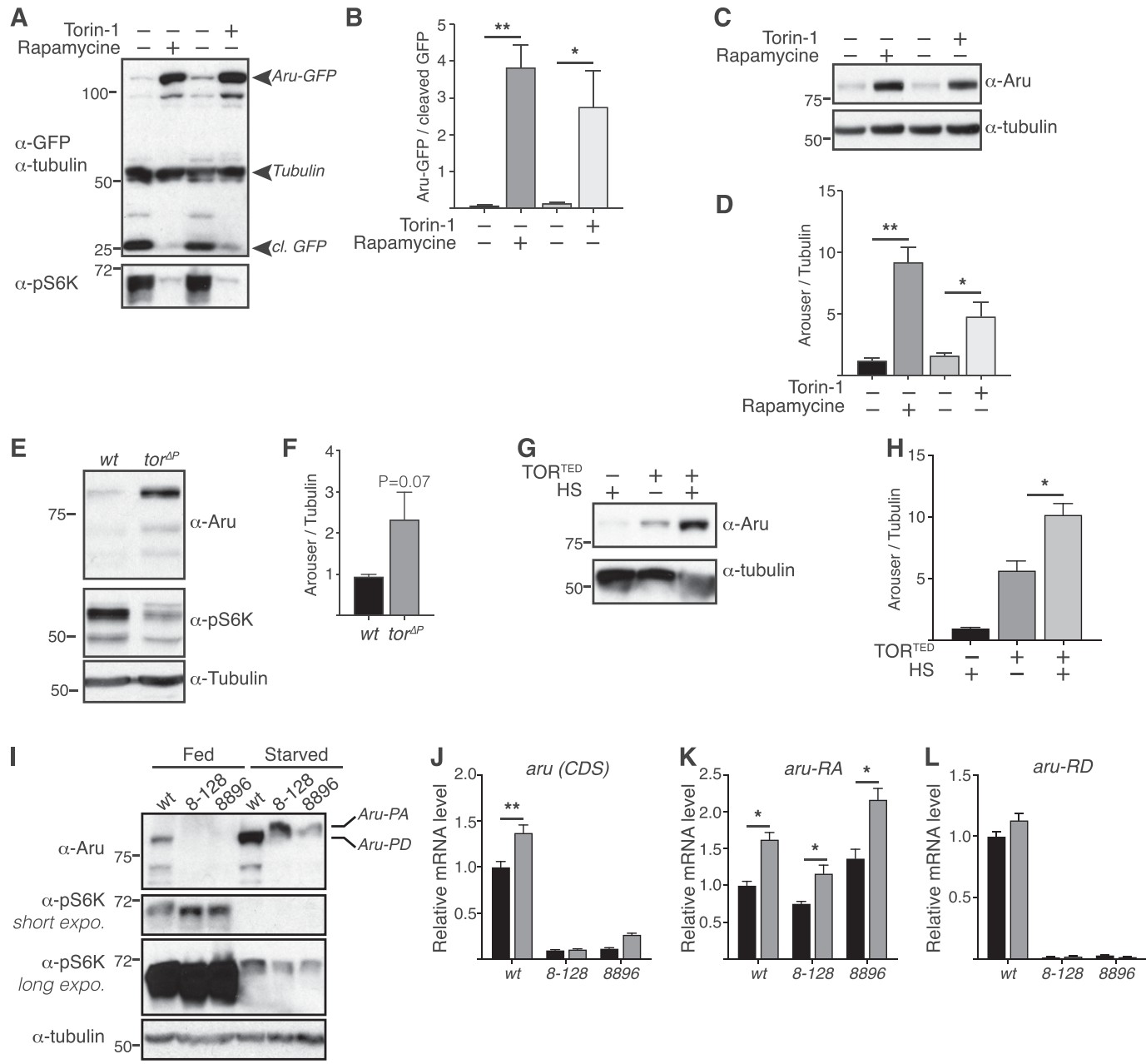

**Figure 3. Arouser functions downstream mTOR.**
**(A, B, C, D)** Western blot analysis and quantification of overexpressed Arouser-GFP (A, B) and endogenous Arouser (C, D) protein level in larvae fed for 24 h with Torin-1 or rapamycin supplemented food. **(E, F)** Western blot analysis and quantification of Arouser protein in larvae lacking mTOR (*tor$^{\Delta P}$*). **(G, H)** Western blot analysis and quantification of Arouser protein level in larvae transiently expressing a kinase dead version of TOR (TOR$^{TED}$) following heat shock (HS). **(I)** Western blot analysis of phosphorylated-S6K (p-S6K) in fed and 24 h starved wild-type (wt) and *aru$^{8-128}$* and *aru$^{8896}$* mutant larvae. **(J, K, L)** Relative *aru* gene expression in fed and 24 h starved larvae using primer sets that recognise either both *aru* isoforms (J), or are specific of *aru-RA* (K) and *aru-RD* (L) isoforms. Bar charts show means ± s.d. Statistical significance was determined using two-tailed *t* test and based on at l–east three independent biological replicates, *P < 0.05, **P < 0.01. Survival experiments show the death rate of at least 100 age- and gender-matched flies per condition. Statistical significance of fly survival was calculated using a Gehan–Breslow–Wilcoxon test.

## mTOR activity is involved in stabilisation of arouser protein

Because Arouser is an adaptor protein that is known to regulate signal transduction pathways, we next sought to decipher which pathways it might regulate under conditions of nutrient deprivation. One major regulator of response to starvation in both

*Drosophila* and mammals is the serine/threonine kinase target of rapamycin (mTOR), which integrates signals from growth factors and nutrient signalling pathways (Scott et al, 2004; Saxton & Sabatini, 2017). Because the nutrient status of the cell/organism serves as a switch for mTOR kinase activity, we first tested the effect mTOR inhibitors on Arouser stability. We observed that Arouser-GFP

(Fig 3A and B) and wild-type (Fig 3C and D) larvae fed for 24 h with food supplemented with mTOR inhibitors Rapamycin or Torin-1 exhibit an elevated level of ectopic Arouser-GFP and endogenous Arouser protein, respectively, suggesting that mTOR activity is required for Arouser degradation. Supporting this, we also observed Arouser accumulation in larvae heterozygote for *Tor* mutation (*Tor*[ΔP/+]) (Fig 3E and F). Therefore, our results suggest that mTOR activity is involved in degradation of Arouser. As the function of mTOR in signalling depends on its kinase activity, we transiently expressed a kinase dead TOR (TOR[TED]) construct in larvae. We observed an accumulation of Arouser protein when TOR[TED] expression was induced by heat shock (Fig 3G and H). Moreover, mass spectrometry analysis of Arouser revealed the presence of phosphorylated residues (Fig S5). These data suggest that Arouser accumulation is related to the kinase activity of TOR, and that it is also phosphorylated.

To assess whether Arouser acts downstream of mTOR, we used the phosphorylation of the S6K as a readout. Active S6K is phosphorylated in fed condition and dephosphorylated during fasting periods (Burnett et al, 1998; Zhang et al, 2000; Kim & Neufeld, 2015). We followed the changes in S6K phosphorylation is fed and 24-h starved *aru* larvae compared with age-matched wild-type larvae. We observed a slight down shifting of the band corresponding to phosphorylated S6K in fed and starved *aru* mutant, suggesting that Arouser is might be involved in S6K signalling following prolonged starvation periods (Fig 3I). Furthermore, at 24-h starvation, we observed for both *aru*-deficient larvae the accumulation of a higher molecular weight band on Arouser probed membrane. This new protein, absent in wild-type lysates, may correspond to the expression of the second isoform Aru-PA under prolonged starvation when the main isoform Aru-PD is absent as we observed a significant increase in the transcript level for *aru-RA* but not *aru-RD* after 24-h starvation (Fig 3J–L).

In summary, our results suggest that the mTOR complex is involved in the stabilisation of Arouser protein in response to starvation, and that Arouser could impact downstream S6K-dependent signalling during prolonged starvation.

## Arouser is involved in insulin signalling

We next investigated Arouser role in insulin signalling as mTOR signalling is central for insulin-mediated signalling events (Harrington et al, 2004; Saxton & Sabatini, 2017) and the integration of insulin signalling is an important component of the adaptation to food availability.

Notably, increased expression of the fat body-derived *Drosophila* insulin peptide 6 (DILP6) is essential to promoting lipid turnover during fasting periods and increasing starvation resistance (Chatterjee et al, 2014). The increased expression of DILP6 has been associated with a reduced expression of the brain insulin peptide DILP2 (Bai et al, 2012). Interestingly, we observed that starved *arouser* mutant larvae failed to up-regulate expression of *dilp6*, which in turn resulted in an unchanged expression in *dilp2* (Fig 4A and B), suggesting that Arouser is may regulate the balance of *dilp2* and *dilp6* expression in response to starvation.

Given the putative roles of Arouser in the regulation of DILP6 signalling, we next sought to expand our analyses to other starvation relevant pathways.

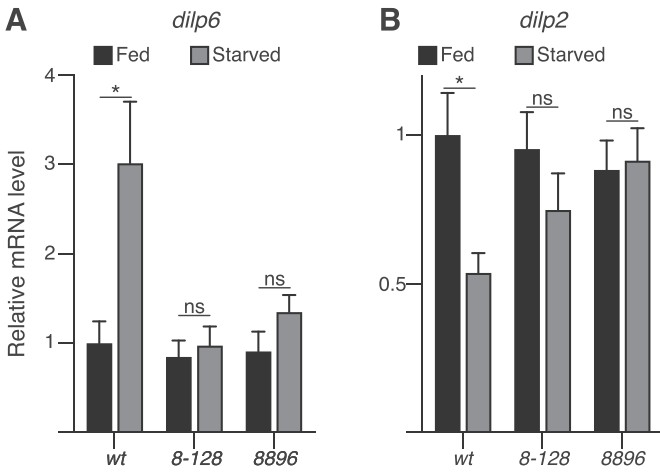

**Figure 4. Arouser is associated with insulin signalling.**
**(A, B)** Relative gene expression for *dilp6* (A) and *dilp2* (B) in fed and 24 h starved wild-type and aru mutant larvae. Bar chart shows means ± s.d. Statistical significance was determined using one-way ANOVA and based on triplicates, *$P < 0.05$, **$P < 0.01$, ***$P < 0.001$.

## Arouser is required for lipid metabolism

A greater sensitivity to starvation may reflect an inability to store fat. In *Drosophila*, fat is stored as lipid droplets in the fat body. Inert fat staining of larval and adult fat bodies revealed that *aru* mutants have significantly smaller lipid droplets than wild-type animals (25–50% smaller) (Fig 5A–D), suggesting that their ability to make large lipid droplets is impaired. Previous work has shown that a high-sugar diet (HSD) increases lipid droplet size in wild-type larvae (Musselman et al, 2011). Interestingly, whereas we could observe an increase in the size of the lipid droplets in control larvae grown on HSD compared with normal diet (ND), no difference was detected in the fat body from *aru* mutants (Fig S6). Moreover, lipid quantification using a colorimetric assay revealed that both *aru* adult mutant flies have a deficit in systemic triglycerides (TAG) (Fig 5E). These observations suggest that Arouser is normally required for droplet size and TAG storage.

To test if Arouser regulates fat metabolism pathways, we compared the expression of key genes related to fat storage and mobilisation in 4-h amino acid–deprived wild-type and *aru* mutant larvae. Interestingly, the relative expression level of some genes related to fat mobilisation and breakdown, such as *bmm*, *AkhR*, and *mcad*, were significantly up-regulated in *aru*-deficient starved larvae (Fig 5F). In addition, some genes linked to fat synthesis and storage, such as *SREBP*, *desat1*, *Acc*, *fas*, *lpin*, *lsd-2*, and *CdsA* were significantly down-regulated (Fig 5G).

To further validate that the changes in lipid metabolism observed in *aru* mutant flies are related to the specific loss of Arouser protein, we use our rescue line that allows for the ectopic expression of Arouser-GFP in *aru*[8-128] mutant background (Fig 5H–K). We observed that both the size of the lipid droplets (Fig 5H and I) and the level of TAG was rescued in adult flies which express Arouser-GFP compared with the deficient line (Fig 5J). Similarly, the expression of genes related to lipid mobilisation was down-regulated in the rescue line, whereas genes associated with lipogenesis were up-regulated (Fig 5K).

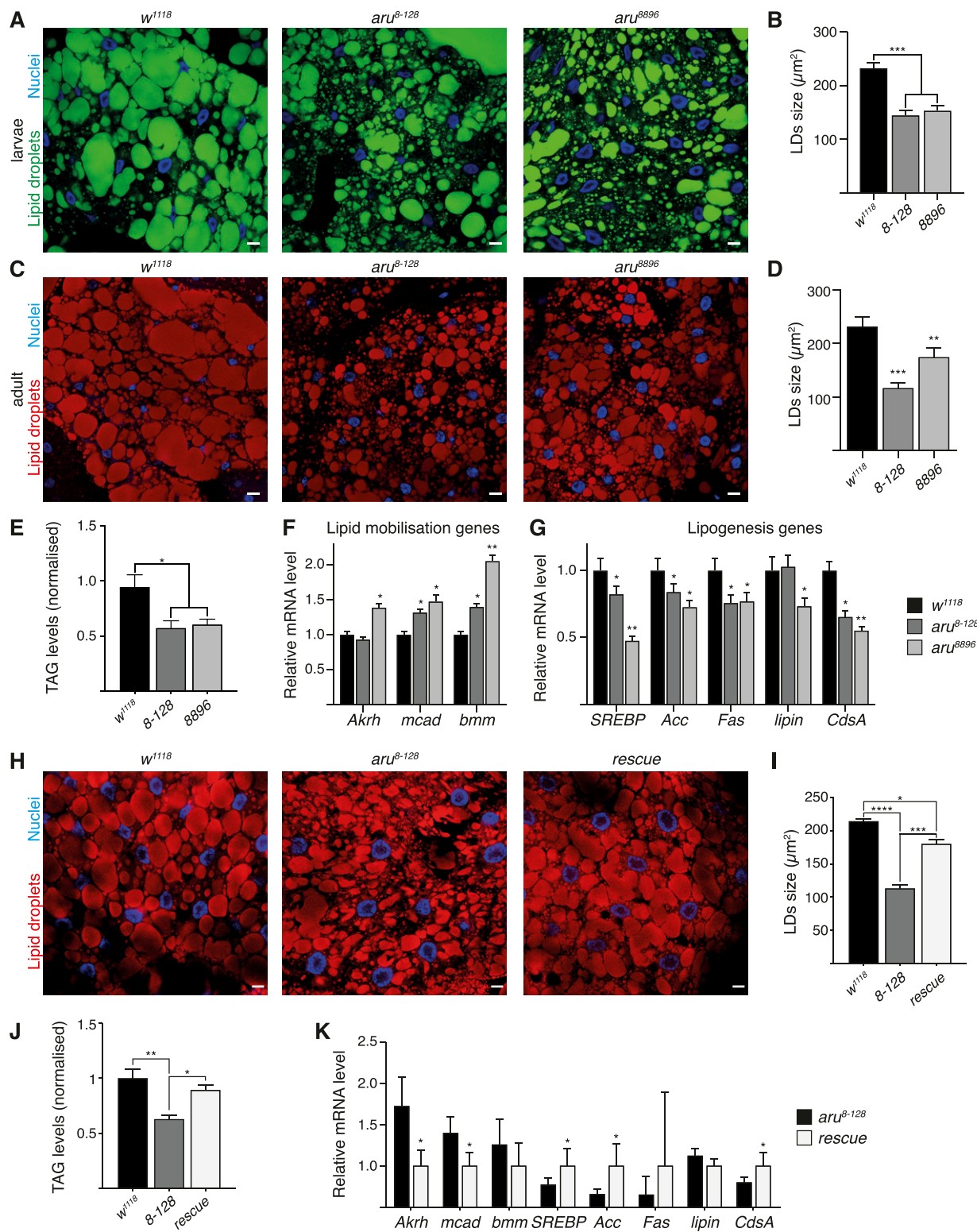

**Figure 5. Arouser-deficient flies have a deficit in lipid.**
**(A, B, C, D)** Confocal section and quantification of the size of the lipid droplets in the fat bodies from fed wild-type and *aru* mutant larvae (A, B) and adult males (C, D) stained with Bodipy (A) or Oil Red O (C). **(E)** TAG quantification in adult wilt-type and *aru* mutant males. **(F, G)** Analysis of relative mRNA levels for genes involved in lipid mobilisation (F) and storage (G) in 4 h starved larvae. **(H, I)** Confocal section and quantification of the size of the lipid droplets in the fat bodies from wild-type, *aru* mutant and rescue larvae. Tissues were stained with Oil Red O (red) and Hoechst (blue). **(J)** TAG quantification in fed adult *aru* rescued males. **(K)** Analysis of relative mRNA levels for genes involved in lipid metabolism in *aru* mutant and rescued 4 h starved larvae. Bar charts show means ± s.d. Statistical significance was determined using one-way ANOVA and based on at least three independent biological replicates, *P < 0.05, **P < 0.01, ***P < 0.001. For microscopy analysis, tissues from at least 10 animals from three independent biological replicates were analysed.

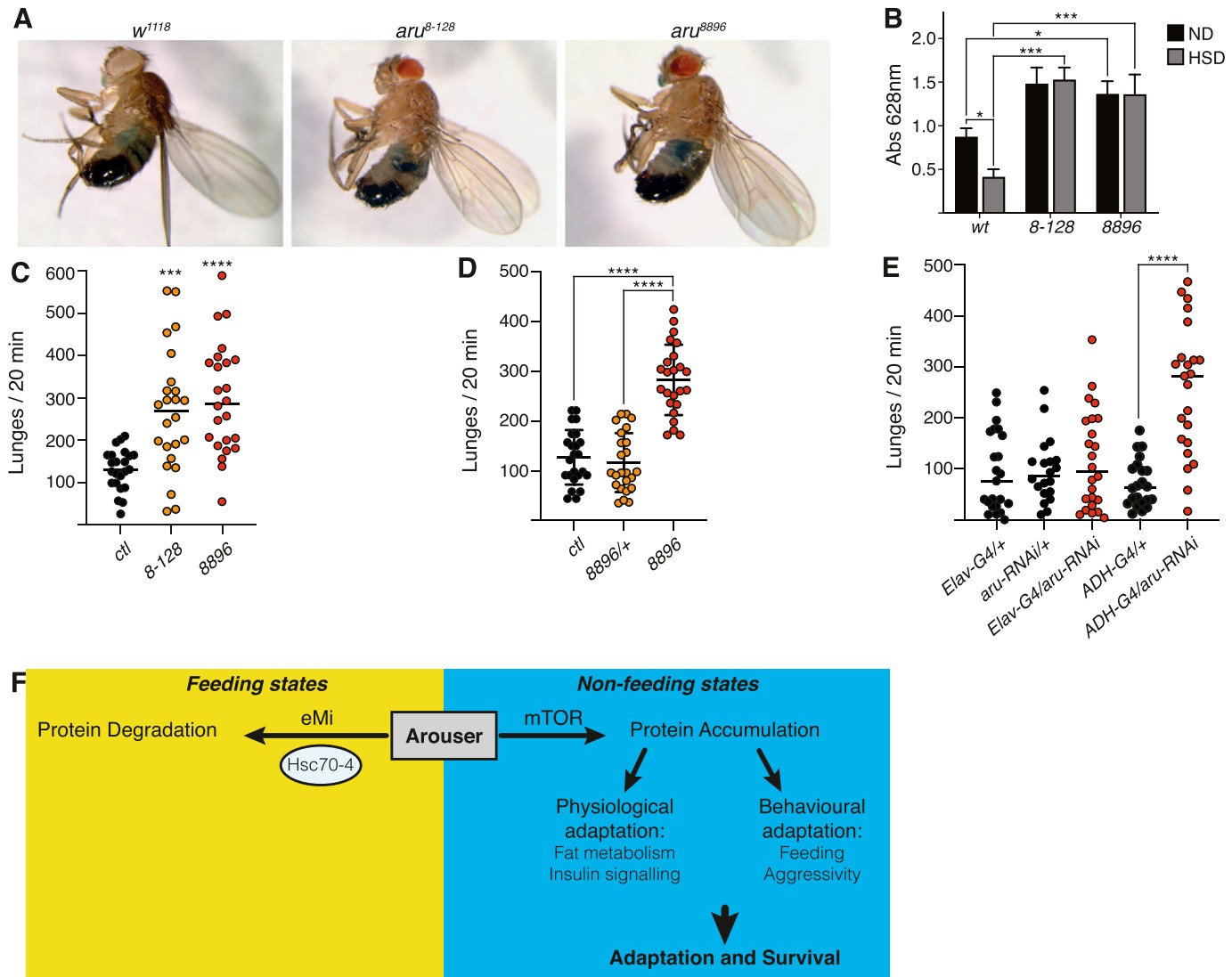

**Figure 6. Feeding, activity and aggressiveness behaviours are affected in *aru*-deficient flies.**
**(A, B)** Relative quantification of the amount of food ingested by wild-type (wt) and aru mutant adult males fed on normal or high sugar diet. **(A)** Representative picture of flies after ingestion of coloured food are shown in (A). Bar chart shows means ± s.d. Statistical significance was determined using two-way ANOVA and based on the analysis of 20–40 individuals, *$P < 0.05$, **$P < 0.01$, ***$P < 0.001$. **(C)** Comparison of the number of lunges per 20 min for *aru* mutants. **(D)** Comparison of the number of lunges per 20 min heterozygote and homozygote *aru^8896^* mutants. **(E)** Comparison of the number of lunges per 20 min of flies silenced for *aru* specifically in the brain (Elav-G4) or fat body (ADH-G4). For the analysis of aggression phenotype **(C, D, E)**, all the individual recordings are shown on charts. **(F)** Model of the balance between degradation and stabilisation of Arouser, and it's implication in fly adaptation and survival during non-feeding states.

Because Arouser protein level is regulated by endosomal micro-autophagy, we evaluated the impact of the loss of the chaperone Hsc70-4 on the size of the lipid droplets. We observed no significant difference in the size of the lipid droplets in fat body cells between *hsc70-4*–deficient larvae, larvae with a rescued *hsc70-4* expression (rescue) and wild-type larvae (Fig S7). However, the overall area of the lipid droplets appeared higher in larvae bearing *hsc70-4* mutation compared with wild-type tissue. Suggesting that eMi might be involved in lipid degradation in *Drosophila* fat body cells as it has been observed in yeast, or in mammals by CMA (Kaushik & Cuervo, 2015; Vevea et al, 2015). The overexpression of Flag-Arouser in the fat body showed a phenotype similar to *hsc70-4* mutant with regard to LDs staining (Fig S7).

In summary we conclude that Arouser is involved in the regulation on fat storage and its mobilisation in the fat body.

### Arouser inhibits feeding and aggression

Because our evidence implicates Arouser in fat metabolism, we next sought to investigate if diet affected the feeding behaviour of *aru* mutants, which have a deficit of TAG reserves. For 5 d, we fed adult males on either a ND or HSD within 24 h from hatching. Food consumption was tested by transferring them onto blue-dyed food for 1 h and measuring the amount of food ingested. As expected, being more calorie dense, we observed that wild-type flies ate less HSD than ND. Arouser-mutant flies, however, ingested significantly

more food when fed on either ND or HSD (Fig 6A and B), suggesting a lack of satiety, even with a higher caloric intake. No difference in the quantity of ingested food was observed for *aru* mutants between HSD and ND (Fig 6B), suggesting that *aru* deficient flies are either unable to sense the changes in dietary sugar or that increasing their caloric intake is not sufficient to rescue their TAG stores. Overall, these results suggest that *aru* mutants have a deficit in regulating energy reserves which is consistent with their increased sensitivity to starvation and the suggested role Arouser plays in lipid mobilisation and breakdown.

Male fruit flies are "territorial" and exhibit an aggressive be-haviour when another fly remains in its food perimeter (Lim et al, 2014). Because the *aru* mutants ingested more food, we wondered if the *aru* mutants might also be more aggressive. Interestingly, *aru* mutant males were hyperaggressive compared with heterozygotes and control flies (Fig 6C and D). Previous data showed that Arouser functions in neurons to regulate behaviour (Eddison et al, 2011; LaFerriere et al, 2011); however, our data demonstrate that Arouser also functions in the fat body. To determine in which cell type Arouser functions to regulate aggression, we knocked down *aru* expression by driving a *UAS-Arouser-RNAi* construct (Eddison et al, 2011) with either *elav*-GAL4 or *ADH*-GAL4 that target neurons or the fat-body respectively. Interestingly, we observed that *aru* knock-down in neurons had no effect on aggression, whereas fat body-targeted knock-down of *aru*, resulted in a significant increase aggression, similar to that seen in the *aru* mutant (Fig 6E). These data suggest that Arouser normally functions in the fat body to regulate aggression and that imbalances in lipid metabolism affect both feeding and aggressive behaviour.

## Discussion

In the present study, we identified Arouser as a novel endosomal microautophagy substrate, and we provide evidence that suggest its involvement in the regulation of lipid metabolism in the fat body. We showed that, in well-fed, nutrient-rich conditions, Arouser is constitutively expressed and degraded by eMi, likely by its tar-geting to the lysosome via its KFERQ motif. However, under nutrient deprivation, we showed that Arouser protein is stabilized. The sta-bilization of Arouser during non-feeding states appears to be im-portant for resistance to starvation as *aru* mutant flies die faster when starved.

We propose a working model where an adaptor protein from the EPS8 proteins family, Arouser, may act at the level of the lysosome, downstream of mTOR to regulate insulin-associated signalling from the fat body. Our data suggest Arouser is required for the survival of the fly during non-feeding states (development and acute star-vation) and promotes the utilization of fat stored in the fat body to produce energy. Although eMi is induced by starvation (Mukherjee et al, 2016; Mejlvang et al, 2018), we show here that basal activity of eMi, in nutrient rich conditions, contributes to the continuous degradation of Arouser protein. Although we did not observe accumulation of Arouser in Atg8a and Atg7 mutants, we cannot exclude the involvement of canonical macroautophagy in the degradation of Arouser. Stabilisation and prevention of Arouser

degradation during starvation is involved in the regulation of lipid metabolism, insulin signalling, and behaviour of the fly (feeding and aggression), processes essential to adaptation and survival (Fig 6F).

Altogether, our data show that endosomal microautophagy in *Drosophila* contributes to metabolic homeostasis through the degradation of the signalling molecule Arouser.

### Arouser effects fat metabolism through signalling pathways

Insulin signalling plays an essential role in the regulation of cell and organismal growth as well as metabolism. In *Drosophila*, seven insulin-like peptides (DILPs) work in concert to regulate growth. During non-feeding states, the fat body specific insulin-like peptide 6 (DILP6) is expressed and secreted into the haemolymph and represses the expression of brain insulin DILP2. Reduced DILP6 expression results in a growth deficit, acute sensitivity to starvation, and reduced lipid stores (Slaidina et al, 2009). These phenotypes are similar to those affecting *aru* flies and therefore suggest that the two genes may be linked. Indeed, we observed that starved *aru* flies fail at inducing *dilp6* expression. Insulin signalling is tightly linked to mTOR signalling as well as response and resistance to starvation. The localisation of Arouser at the lysosome, even in starved conditions, when not being degraded, could imply that its function is located at the lysosomal membrane, where the mTOR complex signals from Puertollano (2014). Indeed, we observed that downstream target genes implicated in lipid metabolism, such as the master regulator SREBP, are down-regulated in *aru* mutant larvae. The down-regulation of the SREBP transcription factors may in turn impair the expression of genes involved in the utilization of fat stores during non-feeding states and starvation.

### Endosomal microautophagy regulates fat metabolism by degrading arouser

Autophagy has recently emerged as a selective degradation pro-cess that allows for the mobilisation of lipid stores in a process call "lipophagy." Indeed, several independent studies have demon-strated a role for autophagy in the turnover of lipid droplets in a broad range of cell types in mammals, yeast, nematodes, and *Drosophila* (Lapierre et al, 2011; Wang et al, 2012; van Zutphen et al, 2014; Schulze et al, 2017). In the condition of nutrient sufficiency, cells store energy in the form of neutral lipids in lipid droplets. These lipid droplets can subsequently be rapidly depleted when nutrients are scarce. The current knowledge of autophagy in lipid metabolism points to lipophagy, which involves specific breakdown of lipid droplets in the lysosome to provide energy under starvation (Singh & Cuervo, 2012). However, little is known about the role of autophagy in the regulation of proteins involved in the signalling related to lipid metabolism. Lipid droplets are formed limited a monolayer of phospholipids and family of lipid droplet coat pro-teins known as perilipins (PLINs) (Beller et al, 2010; Kimmel & Sztalryd, 2016). During starvation, the selective degradation of PLINs by CMA constitutes a prerequisite to increased lipophagy (Kaushik & Cuervo, 2015, 2016). The targeting of proteins for deg-radation by autophagy can be regulated by posttranslational modification of the substrate, and recent studies have shown that

phosphorylation and ubiquitination are necessary for the degradation of selected proteins by CMA (Ferreira et al, 2015; Kaushik & Cuervo, 2016). We have shown that Arouser protein accumulation is associated with the kinase activity of mTOR and is phosphorylated on its residue Ser562. It is, therefore, possible that Arouser is a target for mTOR kinase activity.

The loss-of-function of Hsc70-4 showed no significant impact on the size of the lipid droplets. This can be explained by the fact that deficient eMi would lead to the accumulation of Arouser protein, rather than its accentuated degradation. Moreover, lipid droplets in the fat body from larvae overexpression of Flag-Arouser showed a similar structure as in *hsc70-4* mutant tissue. Fat is virtually stored in every cell of the organism; it is therefore possible that deficiency of eMi will have more potent impacts on the lipid stores in other tissues or cell types.

In this work, we propose a novel indirect role for autophagy, in particular eMi, in lipid metabolism, whereby fat signalling molecules such as Arouser are degraded in fed conditions to regulate fat homeostasis and fly behaviour (Fig 6F).

# Materials and Methods

### Fly stocks and maintenance

Flies used in experiments were kept at 25°C and 70% humidity raised on cornmeal-based feed. ND food contains 130 g of sucrose per litre of food, 400 g for the HSD. The following fly stocks were obtained from the Bloomington *Drosophila* stock center: $w^{1118}$ (#3605), *UAS-GFP* (#5430), *Cg-GAL4* (#7011), hs-GAL4 (#2077), Atg1-RNAi (#26731), Atg13-RNAi (#40861), and $TOR^{TED}$ (#7013). Other UAS and reporter lines are *aru*-RNAi (Eddison et al, 2011), HA-Hsc70-4$^{WT}$ and HA-Hsc70-4$^{3KA}$ (Uytterhoeven et al, 2015), FLPout mCherry-Atg8a (Tusco et al, 2017), and FLPout GFP-mCherry-Atg8a (Jacomin et al, 2015). The following mutant lines have been used: $aru^{8.128}$, $aru^{d8896}$ (Eddison et al, 2011), $Atg8a^{KG07569}$, hsc70-4$^{Δ19}$/TM6c, $w^{1118}$;Df(3R)BSC471/TM6c, hsc70-4 genomic rescue $w^{1118}$;Df(3R)BSC471,Hsc70+/TM6c (gift from P. Verstreken), $cathD^1$, $cathD^{24}$, $spin^{P1}$, and $spin^{EP822}$ (gift from K McCall). The *UAS-Arouser-GFP* transgenic line has been generated using standard injection procedures. The transgenic lines UAS-Flag-Arouser-WT and UAS-Flag-Arouser-AA were generated by cloning the cDNA of aru-RD isoform into pUAST-attB plasmid with addition of a Flag tag in 5′ of *arouser* sequence, the constructs were inserted in the genome using PhiC31-mediated integration (BestGene) using the attP40 docking site. The Arouser rescue line ($aru^{8-128}$ UAS-Aru-GFP) was generated by homologous recombination.

Genotype of the flies used for experiments are listed in Table S2.

### Drug feeding

Second instar larvae were transferred into Nutri-Fly Instant *Drosophila* Medium (66-117; Genesee Scientific) prepared in water supplemented with 2.5–10 $\mu$M chloroquine diphosphate (CQ), 50 $\mu$M rapamycin, 5 $\mu$M Torin-1, or with the drug vehicle only (water for CQ,

0.2% or 0.01% DMSO for rapamycin and Torin-1, respectively). Larvae were kept at 25°C and frozen 24 h later.

### Protein extraction, Western blot analyses, and GST pull-down

Protein content was extracted from early third instar larvae or adult flies in phospholysis buffer (1% Igepal, 50 mM Tris, 120 mM NaCl, 1 mM Na$_3$VO$_4$, 50 mM NaF, 15 mM Na$_4$P$_2$O$_7$, 1 mM benzamidine, 1 mM EDTA, and 6 mM EGTA, pH 6.8; supplemented with Roche cOmplete Mini EDTA-free proteases inhibitor cocktail) using motorised mortar and pestles. Co-immunoprecipitations were performed on lysates from wild-type adult flies. After a 30-min pre-clear of the lysates (1 mg total proteins) with sepharose-coupled G bead (Sigma-Aldrich), the co-immunoprecipitation was performed for 2 h at 4°C using an anti-Arouser antibody and fresh sepharose-coupled G-beads. Four consecutive washes with the lysis buffer were performed before suspension of the beads in 60 $\mu$l 2× Laemmli loading buffer. Protein samples in Laemmli loading buffer were heated for 10 min at 80°C. 10–40 $\mu$g of protein extract or 20 $\mu$l of immunoprecipitation eluates were loaded on acrylamide gels and were transferred onto either nitrocellulose or PVDF membranes (cold wet transfer in 10–20% ethanol for 1 h at 100 V). Membranes were blocked in 5% non-fat milk in TBST (0.1% Tween-20 in TBS) for 1 h. Primary antibodies diluted in TBST were incubated overnight at 4°C or for 3 h at room temperature with gentle agitation. HRP-coupled secondary antibody binding was performed at room temperature for 45 min in 1% non-fat milk dissolved in TBST and ECL mix incubation for 2 min. All washes were performed for 10 min in TBST at room temperature.

The following primary antibodies were used: anti-Arouser (1:1,000) (Eddison et al, 2011), anti-GABARAP (CST, 1:1,000), anti-GFP (sc-9996, 1:1,000; Santa Cruz), anti-phospho S6K (#9209; Cell Signaling Technology), anti-Ref(2)P (ab178440; Abcam), and anti-$\alpha$ tubulin (T5168, 1:40,000; Sigma-Aldrich). HRP-coupled secondary antibodies were from Thermo Fisher Scientific (anti-mouse HRP #31450; anti-rabbit HRP #31460). Following co-immunoprecipitation, Veriblot HRP-coupled IP secondary antibody was used (ab131366, 1:5,000; Abcam).

GST pull-down assays between GST-Atg8a and radiolabelled Arouser were performed as described previously (Tusco et al, 2017).

### Lipid staining and confocal microscopy

Dissected fat bodies from third instar larvae or 1-wk-old adult males were fixed for 30 min in 4% EM grade methanol-free formaldehyde. After three washes in PBS, fat bodies were incubated protected from light in BODIPY 493/503 (D3922, 1:500; Thermo Fisher Scientific) and Hoechst in PBS for 1 h. For Oil Red O staining, fixed tissues were incubated for 30 min in freshly prepared staining solution (3 volumes 0.1% [wt/vol] Oil Red O in isopropanol + 2 volumes water). Washed samples were mounted onto microscope slides in 75% glycerol. All images were acquired using Carl Zeiss LSM880 confocal microscope, using a 63× Apochromat objective. Images were post-processed in Fiji.

### Lipid quantification

Five 10-d-old adult males per genotype homogenised in 300 $\mu$l of 1% Triton X-100 in chloroform. Samples were incubated on ice for 30 min with regular vortexing. After centrifugation for 15 min at 4°C,

14,500g, the chloroform from supernatants containing lipids was evaporated in a vacuum concentrator. The Free Fatty Acid Quantification Colorimetric Assay Kit (ab65341; Abcam) in conjunction with lipase (ab89001; Abcam) was used for quantification of TAG. 200 µl of Assay Buffer was used to resuspend lipids using vortexing, and samples were frozen at −80°C if not immediately quantified.

96-well clear plates with a flat bottom were used for lipid quantification. For TAG quantification, 47 µl of assay buffer was added to a well for each sample, with 2 µl Lipase (ab89001; Abcam) enzyme and 1 µl lipid sample and mixed thoroughly. The rest of the procedure was conducted as directed by the manufacturer (ab65341; Abcam).

### Real-time quantitative PCR

RNA extraction was performed with an Ambion PureLinkTM RNA Mini kit (Thermo Fisher Scientific) according to the manufacturer protocol. 10 larvae were used per extract.

Subsequent steps were performed using 1 µg of total RNA. Thermo Fisher Scientific DNAse I was used to digest genomic DNA. The RevertAid Kit (Thermo Fisher Scientific) was subsequently used to synthesise cDNA. PCR was performed using the GoTaq qPCR Master Mix (A6002; Promega). Primers used for RT-qPCR are listed in Table S3.

### Feeding assay

Adult males were placed within 24 h from hatching either on ND or HSD. After 5 d, the flies were transferred onto ND or HSD food dyed with 2.5% (wt/vol) FD&C Blue #1 for 1 h. Flies were then individually frozen, and lysates were prepared with 50 µl of 0.1% Triton X-100 in PBS. Absorbance was read at 628 nm.

### Lifespan and survival

To measure lifespan and survival in adult flies, males only were collected within 24 h from hatching and cohorts of 20–25 flies were maintained on regular feed at 25°C in a humidified incubator. Three- to 5-d-old males kept on ND were transferred onto 0.7% agar pads made in water only or 5% sucrose solution. Flies were transferred into new tubes every day. Dead events were recorded daily, up to every 2–4 h.

Larval resistance to starvation experiments were conduct by placing well fed staged second instar larvae onto 0.5% agar pads made in water only. The pupae formed were numbered.

### Aggression behaviour analysis

Aggression was measured by recording two flies of the same genotype for 20 min in a 10-mm arena in a 12-cell chamber at 25°C. The bottom of the chamber was coated with a thin layer of apple juice-sugar-agar medium to induce fighting and walls of the chamber coated with Fluon (BioQuip). Flies were habituated to the chamber for 5 min before recording. Movies were recorded at 30 frames per second using gVision (http://gvision-hhmi.sourceforge.net) video acquisition software run with MATLAB (Mathworks). Pairs of males were tracked and automated scores of lunging were derived using

CADABRA software (Dankert et al, 2009). The control for aggression was an independent P element line that had a normal mean/median lunge number (~100 lunges/20 min) compared with a population of 1,600 strains that were initially screened (Eddison M, unpublished).

### In silico analysis of KFERQ-like motifs

The *Drosophila* proteome was screened for protein entries that comprise either of the two KFREQ-like motifs, [KR][FILV][DE][KRFILV]Q and Q[KR][FILV][DE][KRFILV] (Dice, 1990), using the SLiMSearch tool (Krystkowiak & Davey, 2017). Lists were exported into Microsoft Office 365 Excel and hits containing motif sequences were converted into FASTA format for each gene. Protein accessions were converted to gene converted IDs using FlyBase (Thurmond et al, 2019), and these IDs analysed by the Protein Analysis Through Evolutionary Relationships (PANTHER) classification system (pantherdb.org) for molecular function, biological process, cellular component, protein class, and pathway (Mi et al, 2019). Statistical overrepresentation tests were also carried out for molecular functions, biological processes, and cellular components using Fisher's Exact test and False Discovery Rate correction ($P < 0.05$). The data are in Table S4.

### Analysis of arouser phosphosites by LC–MS/MS

Immunoprecipitated endogenous Arouser or overexpressed Arouser-GFP protein were immunoprecipitated from whole larvae before in-gel trypsin digestion. An aliquot containing 20 µl of extracted peptides (from a total of 40 µl) was analysed by means of nanoLC-ESI-MS/MS using the Ultimate 3000/Orbitrap Fusion instrumentation (Thermo Fisher Scientific) using a 90-min LC separation on a 50-cm column.

### Statistical analyses

Statistical analyses were performed using Prism 7 (GraphPad) on at least three independent biological replicates. For the comparison of two groups, *t* test was used. To compare three or more groups, one-way ANOVA with Dunnett's test correction was used. Statistical significance of fly survival was calculated using a Gehan–Breslow–Wilcoxon test. *P*-values are listed in Table S4.

## Data Availability

The mass spectrometry proteomics data have been deposited to the ProteomeXchange Consortium (http://proteomecentral.proteomexchange.org) via the PRIDE partner repository (Perez-Riverol et al, 2019) with the dataset identifier PXD022818.

## Supplementary Information

# Acknowledgements

We thank Dr J Bischof for sending the pUAST-attB plasmid and M Ward and A Torok for fly food preparation. We would like to acknowledge the contribution of the Warwickshire Private Hospital Proteomics Research Technology Platform, Gibbet Hill Road, University of Warwick, UK. The Bloomington *Drosophila* Stock Center contributed to this work by providing mutant and transgenic fly strains. We acknowledge Bestgene Inc for the injection and selection of transgenic flies. This work was supported by Biotechnology and Biological Sciences Research Council grants BB/L006324/1 and BB/P007856/1 awarded to IP Nezis.

## Author Contributions

A-C Jacomin: conceptualization, data curation, formal analysis, validation, investigation, visualization, methodology, and writing—original draft, review, and editing.
R Gohel: data curation, formal analysis, investigation, and methodology.
Z Hussain: data curation, formal analysis, investigation, and methodology.
A Varga: data curation, formal analysis, investigation, and methodology.
T Maruzs: data curation, formal analysis, validation, investigation, visualization, and methodology.
M Eddison: resources, data curation, investigation, and writing—review and editing.
M Sica: data curation, formal analysis, investigation, visualization, and methodology.
A Jain: data curation, formal analysis, investigation, visualization, and methodology.
KG Moffat: resources, data curation, and writing—review and editing.
T Johansen: resources, supervision, investigation, and writing—review and editing.
A Jenny: supervision, investigation, and writing—review and editing.
G Juhasz: data curation, supervision, investigation, visualization, and writing—review and editing.
IP Nezis: conceptualization, data curation, supervision, funding acquisition, visualization, methodology, and writing—review and editing.

## Conflict of Interest Statement

The authors declare that they have no conflict of interest.

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
