## [Reviewer comments · Life Science Alliance]

Life Science Alliance

Degradation of Arouser by endosomal microautophagy is essential for adaptation to starvation

Anne-Claire Jacomin, Raksha Gohel, Zunoon Hussain, Agnes Varga, Tamas Maruzs, Mark Eddison, Margaux Sica, Ashish Jain, Kevin Moffat, Terje Johansen, Andreas Jenny, Gabor Juhasz, and Ioannis Nezis

DOI: <https://doi.org/10.26508/lsa.202000965>

Corresponding author(s): Ioannis Nezis, University of Warwick and Anne-Claire Jacomin

Review Timeline:

Submission Date:	2020-11-18
Editorial Decision:	2020-11-20
Revision Received:	2020-11-20
Editorial Decision:	2020-11-23
Revision Received:	2020-12-01
Accepted:	2020-12-02

Scientific Editor: Shachi Bhatt

Transaction Report:

Please note that the manuscript was previously reviewed at another journal and the reports were taken into account in the decision-making process at Life Science Alliance.

Referee #1:

Jacomin et al., have utilized fantastic in silico screening tools on the *Drosophila* proteome to identify proteins that contain KFERQ-like motifs. They identified these proteins (bearing KFERQ-like motifs) and they were categorized for binding and catalytic activities, biological, cellular and metabolic processes as well as for nucleic acid binding and enzyme modulator features amongst many other biological functions. Arouser was one of the proteins in this screen. It was previously shown to be involved in ethanol sensitivity, memory as well as signal transduction. The authors now provide novel insight on the role of Arouser in regulating lipid metabolism in the fat body of *Drosophila* and further attempt to characterize it as a substrate for endosomal microautophagy. The authors have provided multiple convincing data but several questions remain that I hope can be addressed in a revised manuscript:

Major Comments

- The authors carried out a comprehensive screening as well as utilized effective data extraction tools from the screen. However, the authors do not justify why Arouser was selected from the screen. There are 63 genes for the first motif and 93 for the second. What criteria was used to select Arouser from the list? Please justify...
- Homozygous P-element mutants are used. To show that the phenotypes observed are related to arouser inactivation it is absolutely critical to include a rescue of arouser mutants for all phenotypes. One can never be careful enough and this is even more important when homozygous mutants are being used; it is also the standard in the field.
- In addition, provide UAS-mTOR rescue data in figure3.
- Only one macroautophagy mutant is tested to show that arouser is not degraded by macroautophagy. Include at least one additional macroautophagy mutant.
- Provide additional imaging method with better spatial resolution eg. SIM, STED, Arysca, confirming Arouser localization to late endosome/lysosome.
- The authors assessed whether Arouser acts downstream of mTOR by testing the phosphorylation of S6K in controls and mutants in fed and starved conditions. They don't observe any difference between mutants and control and conclude that arouser is not acting on this pathway. However, it is still not clear if arouser acts downstream of mTOR because other pathways like cyclinD and 4EBP and arouser mutants treated with mTOR inhibitor are not tested.
- GFP is quenched in lysosomes, the authors should provide imaging data of Aru-GFP quenched in fed conditions (Amino Acid supplementation?) and vice-versa under starvation.
- Statistics are missing for the aggression tests (Fig6C-E). In addition, no quantification is provided for the westerns in Fig1K and FigEV2A. Were the westerns performed on the whole animal - larva/adult, or specifically on the fat body of the animals? Also, provide the image data for photoactivatable mCherry-KFERQ-aru mutants in FigEV2C.

Minor Comments

- No reference in the text to Figure 1M

Referee #2:

The manuscript describes a series of features of the protein arouser. These features, however, are not necessarily connected and are not dissected in terms of mechanistic aspects. Thus the overall picture emerging from the manuscript is that of a descriptive study containing a collection of potentially interesting but fragmentary observations that taken together leave the functional role and the mechanism-of-action of arouser elusive.

Arouser is degraded by endosomal Microautophagy (eMI).

The authors start with the observation that arouser possesses a potential eMI signal and observe that the steady state levels of the protein (under feeding conditions) depends on functional lysosomes. The authors conclude that eMI is responsible for protein homeostasis. However, it is not clear to what extent eMI is operative under feeding conditions, as one of the authors reported that eMI is starvation-inducible and that the localization of KFERQ-bearing proteins with late endosomes/lysosomes occurs only after very prolonged starvation, i.e. longer than 12 hrs (Mukherjee et al. Autophagy 2016). On the contrary, arouser colocalizes with LAMP1 structures at steady state. It is also unclear to what extent the signal RLEVQ is responsible for lysosome degradation of arouser. Indeed, it is not assessed whether the RLEAA mutant, which is present in higher amounts as compared to the WT protein, still colocalizes with lysosomes and is still dependent on lysosomes and Hsc70 for its degradation.

Arouser interacts with Atg8a but this interaction does not mediate degradation by macroautophagy. The meaning behind the observation that arouser can be co-IP/pulled down with ATG8 and can localize with it remains unknown, considering that arouser is not degraded by autophagy. Does arouser regulate autophagy? It is also unclear how the pull down experiment with radiolabelled arouser was performed.

Arouser protein stabilisation depends on mTOR activity.

The authors establish a correlation between mTOR activity and the half-life span of arouser but neither explore the nature of this correlation nor address the obvious question as to whether arouser is a substrate of mTOR.

Arouser is involved in insulin signalling and is required for lipid metabolism

The authors report that arouser mutants show reduced expression of DILP6 and show dysregulation in fat biosynthesis/mobilization, two potentially interesting observations whose underlying molecular mechanisms remain totally unexplored.

Referee #3:

In their study Jacomin et al identified the adaptor protein arouser as a new target for endosomal microautophagy (eMi). The authors also provide evidence for a possible involvement of arouser in regulating fat metabolism in *Drosophila*. Interestingly, arouser cellular levels are being regulated whereby during feed condition it is consciously degraded by eMi while during starvation its degradation is inhibited. A link between arouser and insulin signaling is also provided.

While some of the phenomena described in this study are potentially interesting, the study remains too preliminary to support the many assumptions made by the authors. For example, the first in silico screen to identify proteins harboring KFERQ-like motifs is only generally described, no attempt was made to experimentally verify the different targets. Nevertheless, the authors present these as actual targets for eMi, which is quite misleading. Next, arouser is been selected for further characterization with no clear rationale other than indicating that it is an "adaptor protein" (adaptor for what?).

More crucially, the authors convincingly show that arouser is being degraded in the lysosome, however the conclusion that it is targeted to the lysosome via eMi is not well-based. A more rigorous analysis of the requirements for this process are needed, particularly missing are experiments to show that the degradation is directly mediated by MVBs in ESCRT-dependent manner. Along this line, it is important to test the effect of multiple core Atg proteins to rule out macroautophagy (or a subset pathway of non-canonical autophagy). In addition, there was no attempt to determine whether arouser is delivered to the lysosome by CMA.

The authors conclude that they uncovered a novel role for eMi in the regulation of lipid metabolism during starvation. To validate this hypothesis, it is important to determine whether mutation in the actual eMi machinery also affect this process. Finally, the link to insulin signaling should be better characterized.

Additional comments

Quantifications describe throughout are missing the number of repetitions.

Bars are missing in microscopy images

Fig 1B- Colocalization should be quantified and more importantly, an attempt to localize the endogenous protein should be made.

Fig 1E- in spinEP the Ref (2)P band is lower than the control, needs an explanation.

Fig 1 G-J- consistency is needed in sample naming: WB hsc70, vs graphs hsc70-4.

Fig 1G- positive control for microautophagy is missing.

Fig 1I- in the rescue right lane, there are no bands of arouser.

Fig 1M- Is not described in text.

Fig 2 legend- mismatch between B/C vs results

Fig 2B- describe at which starvation time point mRNA was taken.

Fig 3 explanation for PS6K is missing- why the levels are low after treatment? Additional assays for torin and rapamycin activity should be added. Arouser is expressed differently in A and B, needs to be mentioned in text.

Referee #1 Review

Report for Author:

In this paper, Jacomin et al report Arouser as a novel substrate of endosomal microautophagy (eMi). Additionally, by using Arouser null mutants, Jacomin et al demonstrate a role for Arouser in lipid metabolism in the *Drosophila* fat body and mTOR signaling. While this is a potentially interesting link between eMi and energy homeostasis, the study is very descriptive and how the two pathways are connected was not explored. I think that quite some work is needed:

- In figure 1, the authors show that Arouser is degraded via lysosomes. Please assess, as control,

Arouser protein level when blocking proteasomal degradation and mRNA levels after CQ feeding.

- Could the authors speculate on why during starvation, Arouser localizes with lysosomes (figure EV2A-C) while the protein is accumulating (figure 2A-C)? Could they also suggest how and what is the signal that prevents Arouser from being degraded during starvation?

- Could the authors explain why in the blot shown in figure 2H, endogenous Arouser protein is detected (rescue condition) while no mRNA is present (figure 2I)?

- Based on mass spec results, Jacomin et al suggest Arouser as being a substrate for mTOR-mediated phosphorylation. However, authors should consider the possibility of Arouser being phosphorylated by other kinases or, to rule this out, assess via mass spec the phosphorylation state of Arouser in the presence of wild type and kinase dead mTOR.

- The blot in figure 3I shows the presence of phosphorylated S6K in wild type and aru null flies, but I am not sure the small shift of the band is proof that S6K is not phosphorylated. This needs to be strengthened. The antibody is specific for phosphorylated S6K that, as expected, in starved condition cannot be detected (with short exposure). As an additional control, a blot anti-pS6K could be performed on kinase dead mTOR flies.

Moreover, on the same blot in figure 3I, a higher molecular size band is detected for Arouser. This is hypothesized to be the longer aru-PA transcript. As aru null flies starved for 4h do not show this additional band (figure 2H), this could likely be a compensatory mechanism occurring because the flies were starved for 24h. Therefore, the authors should keep this in mind when interpreting results obtained from flies starved for such a long period. Additionally, figure 3K shows that mRNA of the transcript aru-PA is also detected in starved wild type flies. However, this is not visible on the blot in figure 3I. Could the authors speculate on this? Maybe the authors should consider using an aru knock out flies instead.

- In figure EV3, the authors test the possible involvement of Arouser in macroautophagy. Please provide a positive and a negative control for the IP in figure EV3C. Is the detected Atg8a lipidated or non-lipidated? Given that Arouser is not degraded by macroautophagy and does not have a function in this pathway, could the authors speculate on the possible function of the Atg8a-Arouser interaction?

- From figure EV4 Jacomin et al conclude that macroautophagy is not affected in the fat bodies of Arouser knock down flies. Could the authors provide mRNA level data to show the extent of the knock down of the RNAi line that was used and explain why for this experiment they used an RNAi instead of the null alleles (that were used in the other experiments?). Additionally, the authors should provide quantifications for the images presented and explain why two different tools were used for fed and starved conditions. Moreover, could the authors speculate on why the pattern of LTR is so different between fed and starved conditions?

- In figure 4 the authors link Arouser to dilp6 levels. A rescue experiment should be performed by expressing GFP-Arouser in the null background. Additionally, to further prove the link between Arouser and insulin signaling, the authors should overexpress Arouser in wild types, as a way to mimic starvation when Arouser builds up. In this condition, dilp6 levels should increase while dilp2 levels should decrease.

- In figure 5 the authors propose a function for Arouser in TAG metabolism. To complement the data on LDs size, the authors should provide measurements of overall fat body mass and cell size. To better connect Arouser, Insulin and TAG metabolism, the authors should overexpress dilp6 in Arouser null flies and measure LDs size and TAG level. Moreover, in figure 5K mRNA levels in aru8-128 and rescue should be compared with mRNA level in w1118. Additionally, could the authors speculate on the functional significance of the connection between eMi and lipid metabolism?

- Jacomin et al. raised the possibility of Arouser as being indispensable for satiety perception or TAG storage. The authors should measure TAG level in flies fed with an HSD.

Referee #2 Review

Report for Author:

The authors have satisfactorily addressed the concerns raised in my previous review.

Referee #3 Review

Report for Author:

Jacomin et al. identified Arouser as a new target of endosomal microautophagy (eMI). Interestingly, it is continuously degraded under fed "control" conditions while under starvation it is stabilized.

Evidence for Arouser involvement in the regulation of lipid metabolism and correlated food-related behavior in flies are shown as well as a possible link to TOR and insulin-signaling.

On face value the working hypothesis, linking endosomal degradation of Arouser to energy homeostasis looks rather attractive and novel, however despite the effort made in the revision process, the study remains too preliminary to convincingly support the suggested model. The authors describe a series of experiments describing different phenomena but fail to provide the necessary evidence to signify many of them. For example, the link of Arouser degradation to eMI is still rather weak. The possibility that other forms of autophagy may act be involved was not properly addressed as removal of Atg8a or Atg7 in the flies is not sufficient to exclude canonical autophagy. No attempt was made to determine the mechanism that controls specific degradation of Arouser under control conditions but not under starvation. The later seem extremely important for the suggested model. The suggested link to mTOR does not provide any valuable information, remaining rather descriptive and correlative. The link to insulin signaling too does not help. In addition, the entire chapter devoted to the interaction of Arouser with Atg8a, mostly describe negative data and is rather confusing.

In summary, the authors fail to convincingly address many of the key editorial requests.

The rationale for choosing arouser for further analysis is rather weak.

The experiments done to substantiate eMI are limited.

The link to mTOR remain rather descriptive and the insights to arouser role in insulin signaling and the consequent lipid metabolism are insufficient.

November 20, 2020

Re: Life Science Alliance manuscript #LSA-2020-00965-T

Dr. Ioannis Nezis
University of Warwick
School of Life Sciences
Gibbet Hill Road
Coventry, West Midlands CV4 7AL
United Kingdom

Dear Dr. Nezis,

Thank you for submitting your manuscript entitled "Degradation of Arouser by endosomal microautophagy is essential for adaptation to starvation" to Life Science Alliance (LSA).

For a brief overview, the manuscript was reviewed at two Alliance journals, from where the manuscript and the accompanying reviews were transferred to LSA, with the authors permission. After the second round of review, one of the reviewers' still remained unconvinced of the link between endosomal degradation of Arouser and energy homeostasis, and was particularly concerned about the correlative nature of the link between Arouser and mTOR or insulin signaling, and the exclusion of canonical autophagy. After reviewing all the reviewer reports from previous journals, LSA editors were intrigued by the findings provided a novel insight on the role of Arouser in regulating lipid metabolism, and further characterized Arouser as a substrate for endosomal microautophagy. LSA editors deemed that these findings were interesting enough that the study can be published at LSA, provided the authors address the caveats of mechanisms proposed. The following minor textual changes are requested,

- + please edit the manuscript discussion to clarify that the removal of Atg8a or Atg7 in flies may not sufficiently exclude canonical autophagy (as pointed out by Rev 3, 2nd round of review)
- + please tone down the connection to mTOR and insulin signaling (pointed out by Rev 3, 2nd round of review)
- + please improve the presentation of data pertaining to the interaction (or lack of) between Atg8a and Arouser, for clarity
- + please provide a point-by-point response to the Rev 3's concerns

We should also note that the attached review by Rev 1 was in error, as the reviewer later realized that this was a revised manuscript, after which s/he signed off that the authors addressed all their concerns.

Thank you for this interesting contribution to Life Science Alliance. We are looking forward to receiving your revised manuscript.

Sincerely,

Shachi Bhatt, Ph.D.
Executive Editor
Life Science Alliance
<https://www.lsajournal.org/>
Tweet @SciBhatt @LSAJournal

- A letter addressing the reviewers' comments point by point.
- An editable version of the final text (.DOC or .DOCX) is needed for copyediting (no PDFs).
- High-resolution figure, supplementary figure and video files uploaded as individual files: See our detailed guidelines for preparing your production-ready images, <https://www.life-science-alliance.org/authors>
- Summary blurb (enter in submission system): A short text summarizing in a single sentence the study (max. 200 characters including spaces). This text is used in conjunction with the titles of papers, hence should be informative and complementary to the title and running title. It should describe the context and significance of the findings for a general readership; it should be written in the present tense and refer to the work in the third person. Author names should not be mentioned.

B. MANUSCRIPT ORGANIZATION AND FORMATTING:

We encourage our authors to provide original source data, particularly uncropped/-processed electrophoretic blots and spreadsheets for the main figures of the manuscript. If you would like to

add source data, we would welcome one PDF/Excel-file per figure for this information. These files will be linked online as supplementary "Source Data" files.

Editorial comments

- 1) Please edit the manuscript discussion to clarify that the removal of Atg8a or Atg7 in flies may not sufficiently exclude canonical autophagy (as pointed out by Rev 3, 2nd round of review)
We now clearly mention in the discussion, lines 384-385 that: 'Although we didn't observe accumulation of Arouser in Atg8a and Atg7 mutants, we cannot exclude the involvement of canonical macroautophagy in the degradation of Arouser'.
- 2) please tone down the connection to mTOR and insulin signaling (pointed out by Rev 3, 2nd round of review)
We have toned down the connection to mTOR and insulin signalling throughout the text lines: 214, 252, 253, 254, 267, 272, 285, 298, 299, 378 (highlighted in yellow)
- 3) please improve the presentation of data pertaining to the interaction (or lack of) between Atg8a and Arouser, for clarity
We have improved the presentation related to the interaction between Atg8a and Arouser lines 185-213 (highlighted in yellow)
- 4) please provide a point-by-point response to the Rev 3's concerns

Reviewer 3 comments

The possibility that other forms of autophagy may act be involved was not properly addressed as removal of Atg8a or Atg7 in the flies is not sufficient to exclude canonical autophagy.

We now clearly mention in the discussion, lines 384-385 that: 'Although we didn't observe accumulation of Arouser in Atg8a and Atg7 mutants, we cannot exclude the involvement of canonical macroautophagy in the degradation of Arouser'.

No attempt was made to determine the mechanism that controls specific degradation of Arouser under control conditions but not under starvation. The later seem extremely important for the suggested model.

We have toned down the conclusions in the discussion (highlighted in yellow)

The suggested link to mTOR does not provide any valuable information, remaining rather descriptive and correlative. The link to insulin signaling too does not help.

We have toned down the connection to mTOR and insulin signalling throughout the text lines: 214, 252, 253, 254, 267, 272, 285, 298, 299, 378 (highlighted in yellow)

In addition, the entire chapter devoted to the interaction of Arouser with Atg8a, mostly describe negative data and is rather confusing.

We have improved the presentation related to the interaction between Atg8a and Arouser lines 185-213 (highlighted in yellow)

The rationale for choosing Arouser for further analysis is rather weak.

We chose to carry on our analysis on Arouser as it is a member of the EPS8 proteins family, and that one of them has been suggested to be regulated by chaperone-mediated autophagy (CMA) in mammals (Welsch *et al.*, 2010). Importantly, our screen identified the gene *Comatose/Comt* which was recently validated as a control for eMi in *Drosophila*, thus indirectly validating our screen. We have also added the RT-qPCR validation of the *hsc70-4* mutant, showing that *hsc70-4* expression is reduced, but not *Aru* expression, so the accumulation of Arouser protein in those mutant does not result from an increased gene expression (Figure 1D). We have now updated the manuscript, including extended explanation on Arouser protein (function, EPS8 family protein), and why we decided to investigate it in the context of eMi.

The experiments done to substantiate eMi are limited.

Thanks for this comment. We have added new sets of data in Figure EV2 to reinforce the implication of eMi for the degradation of Arouser (ESCRT RNAi larvae and expression of *Hsc70-4* mutant that lack the ability to deform the membranes). We have also toned down the conclusions in the discussion (highlighted in yellow)

The link to mTOR remain rather descriptive and the insights to Arouser role in insulin signaling and the consequent lipid metabolism are insufficient

We have toned down the connection to mTOR and insulin signalling throughout the text lines: 214, 252, 253, 254, 267, 272, 285, 298, 299, 378 (highlighted in yellow)

November 23, 2020

RE: Life Science Alliance Manuscript #LSA-2020-00965-TR

Dr. Ioannis Nezis
University of Warwick
School of Life Sciences
Gibbet Hill Road
Coventry, West Midlands CV4 7AL
United Kingdom

Dear Dr. Nezis,

Thank you for submitting your revised manuscript entitled "Degradation of Arouser by endosomal microautophagy is essential for adaptation to starvation". We would be happy to publish your paper in Life Science Alliance pending final revisions necessary to meet our formatting guidelines.

Along with the points listed below, please also attend to the following:

- please add ORCID ID for secondary corresponding author-they should have received instructions on how to do so
- please update your EV & appendix figures and rename them as supplementary figures in your figure legends and in your main manuscript text. For more information, please visit this page: <https://www.life-science-alliance.org/manuscript-prep#figsvids>
- please upload your tables as editable doc or excel files and add your table legends to the main manuscript text
- please deposit the mass spec data in a relevant public database and provide the accession number in a 'Data Availability' section in the manuscript (<https://www.life-science-alliance.org/manuscript-prep#datadepot>)
- please specify the category of the manuscript at re-submission
- please add scale bars to the fluorescent images in Figure EV2, EV3
- we encourage you to move the labels out of the panels in Figure EV4

A. FINAL FILES:

B. MANUSCRIPT ORGANIZATION AND FORMATTING:

Sincerely,

Shachi Bhatt, Ph.D.
Executive Editor
Life Science Alliance

<https://www.lsjournal.org/>
Tweet @SciBhatt @LSAJournal

December 2, 2020

RE: Life Science Alliance Manuscript #LSA-2020-00965-TRR

Dr. Ioannis Nezis
University of Warwick
School of Life Sciences
Gibbet Hill Road
Coventry, West Midlands CV4 7AL
United Kingdom

Dear Dr. Nezis,

Thank you for submitting your Research Article entitled "Degradation of Arouser by endosomal microautophagy is essential for adaptation to starvation". It is a pleasure to let you know that your manuscript is now accepted for publication in Life Science Alliance. Congratulations on this interesting work.

DISTRIBUTION OF MATERIALS:

Again, congratulations on a very nice paper. I hope you found the review process to be constructive and are pleased with how the manuscript was handled editorially. We look forward to future exciting submissions from your lab.

Sincerely,

Shachi Bhatt, Ph.D.

Executive Editor

Life Science Alliance

<https://www.lsjournal.org/>
